# SARS-CoV-2 Infections in Vaccinated and Unvaccinated Populations in Camp Lemonnier, Djibouti, from April 2020 to January 2022

**DOI:** 10.3390/v14091918

**Published:** 2022-08-30

**Authors:** Catherine E. Arnold, Logan J. Voegtly, Emily K. Stefanov, Matthew R. Lueder, Andrea E. Luquette, Robin H. Miller, Haven L. Miner, Andrew J. Bennett, Lindsay Glang, Tara N. McGinnis, Kristie E. Reisinger, Jae W. Dugan, Michael A. Mangat, Daniel J. Silberger, Rebecca L. Pavlicek, Chaselynn M. Watters, Gregory K. Rice, Francisco Malagon, Regina Z. Cer, Stephen M. Eggan, Kimberly A. Bishop-Lilly

**Affiliations:** 1Naval Medical Research Center, Fort Detrick, Frederick, MD 21702, USA; 2Defense Threat Reduction Agency, Fort Belvoir, VA 22060, USA; 3Leidos, Reston, VA 20190, USA; 4U.S. Naval Medical Research Unit No. 3 (NAMRU-3), 95030 Sigonella, Italy; 5Expeditionary Medical Facility, CLDJ-HOA, Camp Lemonnier 09622, Djibouti; 6Navy Environmental Preventive Medicine Unit-Two, Norfolk, VA 23511, USA; 7Naval Medical Research Center, Silver Spring, MD 20910, USA

**Keywords:** SARS-CoV-2, bioinformatics, genomics, surveillance, Djibouti

## Abstract

The global pandemic caused by severe acute respiratory syndrome coronavirus 2 (SARS-CoV-2) has highlighted the disparity between developed and developing countries for infectious disease surveillance and the sequencing of pathogen genomes. The majority of SARS-CoV-2 sequences published are from Europe, North America, and Asia. Between April 2020 and January 2022, 795 SARS-CoV-2-positive nares swabs from individuals in the U.S. Navy installation Camp Lemonnier, Djibouti, were collected, sequenced, and analyzed. In this study, we described the results of genomic sequencing and analysis for 589 samples, the first published viral sequences for Djibouti, including 196 cases of vaccine breakthrough infections. This study contributes to the knowledge base of circulating SARS-CoV-2 lineages in the under-sampled country of Djibouti, where only 716 total genome sequences are available at time of publication. Our analysis resulted in the detection of circulating variants of concern, mutations of interest in lineages in which those mutations are not common, and emerging spike mutations.

## 1. Introduction

Severe acute respiratory syndrome coronavirus 2 (SARS-CoV-2) was first identified in Wuhan, China, in December of 2019 [1,2] and causes a severe respiratory disease, termed coronavirus disease 2019 (COVID-19) [3]. The World Health Organization (WHO) declared SARS-CoV-2 a pandemic on 11 March 2020, and the virus caused infections on all continents, with over 533 million cases and over 6.3 million deaths worldwide as of 8 June 2022 [4]. SARS-CoV-2 is a member of the *Coronaviridae* family and has a positive-sense single-stranded (ssRNA(+)) RNA genome that consists of one linear RNA segment. One feature that partly distinguishes coronaviruses from other RNA viruses is the proofreading ability of the virus during transcription. There are 14 open reading frames (ORFs) with ORF1ab, the largest ORF in the genome, encoding a polypeptide [5]. The mature peptide nsp14, which is cleaved from the ORF1ab polypeptide, is an exonuclease that can proofread the nascent RNA being transcribed, excising incorrectly incorporated nucleotides [6]. Although nsp14 performs this important function, mutations still occur, and resulting variants emerge. Due to the nature of the prolonged pandemic, numerous variants have emerged, with some harboring concerning mutations that may affect the efficacy of diagnostic, prophylactic, and therapeutic countermeasures, as well as transmission rates. Examples include the emergence of lineage Alpha (B.1.1.7), a WHO-designated variant of concern (VOC); Alpha is the first lineage with this distinction and was shown to have a higher transmission rate [7]. Subsequent VOCs Beta (B.1.351), Delta (B.1.617.2), and Omicron (B.1.1.519) also have mutations in the receptor-binding domain (RBD) of the spike protein, affecting the ability of antibodies to bind and neutralize the virus [8,9,10,11].

Various technologies exist for the detection of SARS-CoV-2, including polymerase chain reaction (PCR), nanotechnology-based sensors, and viral genome sequencing [12,13]. While PCR is a method very commonly implemented, it provides only limited information (i.e., the presence of the virus or its genes), similar to nanosensors. Conversely, viral genomic sequencing allows for a full analysis, including elucidation of mutations and lineage information as well as tracking the spread and emergence of variants and the associated effectiveness of available countermeasures. SARS-CoV-2 genomic sequencing in African countries has been vastly underrepresented when compared with that of other countries such as the United Kingdom (UK), China, and the United States. Prior to our sequencing efforts, no sequences had been published for Djibouti in the Global Initiative on Sharing Avian Influenza Data (GISAID) [14], a heavily used database containing SARS-CoV-2 sequences. Two and a half years since the first identified SARS-CoV-2 cases, as of June 2022, the surrounding countries had limited sequences published to GISAID (not including the genome submissions reported herein as part of our work): 35 for Somalia, 626 for Ethiopia, and 0 for Eritrea.

Multiple vaccines were developed for SARS-CoV-2, and within the UK, the ChAdOx1 nCoV-19 vaccine AZD1222 (AstraZeneca) received regulatory approval on 2 December 2020 by the UK medicines’ regulator Medicine and Healthcare products Regulatory Agency (MHRA) [15]. Sputnik V (Gamaleya Research Centre) was developed and approved in Russia in February 2020 [16]. CoronaVac COVID-19 vaccine (Sinovac) was developed and approved in China in June 2021 [17]. Within the United States, BNT162b2 (Pfizer-BioNTech) was the first COVID-19 vaccine to gain the U.S. Food and Drug Administration (FDA)’s emergency use authorization (EUA) approval on 11 December 2020, followed by full FDA approval on 23 August 2021 [18,19]. The Moderna (mRNA-1273) COVID-19 vaccine received the FDA’s EUA approval on 18 December 2020 and full approval on 31 January 2022 [20,21]. The Janssen COVID-19 vaccine was EUA-approved on 27 February 2021 [22]. The most recent vaccine approved for FDA’s EUA is the Novavax NVX-CoV2373 COVID-19 vaccine, approved on 13 July 2022 [23]. Both Pfizer-BioNTech and Moderna are two-dose mRNA vaccines, with three weeks and four weeks recommended between doses, respectively. AstraZeneca and Sputnik V are two-dose adenovirus vector vaccines, with a recommended time between doses of 8–12 weeks for AstraZeneca and 21 days for Sputnik V. Sinovac is a two-dose inactivated viral vaccine, administered 2–4 weeks apart. Janssen is a single-dose adenovirus vector vaccine, while Novavax is a two-dose recombinant protein vaccine administered 21 days apart.

For all the vaccines (except the single-dose Janssen vaccine), the recipient is considered fully vaccinated two weeks after the second dose. Instances of infection in fully vaccinated individuals have been documented and are not unexpected, as the maximum efficacy of the vaccines is 95% in those without prior infection (for Pfizer-BioNTech) [18]. Other vaccines, including Sputnik V, Sinovac, AstraZeneca, Janssen, Novavax, and Moderna, had lower than 95% efficacy rates [15,16,17,18,20,22,24]. At the time of vaccine efficacy trials, the VOCs were not widely circulating, but recent studies have shown some reduction in efficacies against the VOCs specifically. Studies in Qatar showed reduced effectiveness of the Pfizer vaccine against three of the VOCs, including Alpha, Beta, and Delta, with Beta and Delta showing the greatest evasion of vaccine-induced immunity [25,26]. Studies show reduced neutralization of Beta and Delta by convalescent and vaccinated sera, but minimal immune evasion by Alpha [8,9,27,28,29]. A study in Scotland showed reduced efficacy of the AstraZeneca vaccine against both Alpha and Delta [30]. Finally, a South African study showed the Pfizer vaccine had reduced efficacy against the latest VOC, Omicron In this study, we described 589 SARS-CoV-2 genomes, including those from 196 cases of vaccine-breakthrough infections (VBI) at Camp Lemonnier, Djibouti, in individuals vaccinated with the Moderna, Pfizer-BioNTech, Janssen, AstraZeneca, Sinovac, or Sputnik V vaccines.

## 2. Materials and Methods

### 2.1. Sample Collection

From April 2020 through January 2022, the 795 patient samples in this study were collected at Camp Lemonnier, Djibouti, and tested positive for SARS-CoV-2. The samples were collected from symptomatic and asymptomatic patients via nasopharyngeal swabs and then placed into viral transport media (VTM). All samples tested SARS-CoV-2 positive via one of the FDA emergency-use-authorized BioFire panels: Respiratory Panel 2.1 or COVID-19 Test v1.1. After testing positive for SARS-CoV-2, all samples were stored at −20 °C for further testing.

### 2.2. RNA Extraction and Genome Sequencing

Viral genome sequencing and analysis was conducted from primary material at Naval Medical Research Center, Fort Detrick, MD, under non-human subject research determination PJT 20-08. Briefly, RNA was extracted from 0.25 mL of VTM using 0.75 mL of TRIzol LS reagent (Invitrogen; Carlsbad, CA, USA) according to the manufacturer’s protocol. The RNA concentration was measured using a Qubit RNA High Sensitivity assay (ThermoFisher Scientific; Waltham, MA, USA) prior to use in the ARTIC nCoV-2019 sequencing protocol [31], the NEBNext ARTIC SARS-CoV-2 Library Prep Kit (New England Biolabs, Ipswich, MA, USA), and/or the QIAseq DIRECT SARS-CoV-2 Kit (Qiagen, Valencia, CA, USA). Briefly, the RNA was reverse-transcribed, and cDNA was then amplified using multiplex PCR and either the associated ARTIC primer pools or the QIAseq DIRECT primer pools. The inserts were then polished and ligated to Illumina-compatible adaptors and indexes. The libraries were quality-checked using an Agilent High Sensitivity DNA kit (Agilent Technologies; Santa Clara, CA) and quantitated using a Qubit DNA High Sensitivity assay (ThermoFisher Scientific) prior to sequencing using Illumina MiSeq v3 2 × 300 sequencing kits and MiSeq sequencers (Illumina; San Diego, CA, USA).

### 2.3. Bioinformatic Analyses

Viral Amplicon Illumina Workflow (VAIW) [32] was used to collate and analyze the SARS-CoV-2 genomes from resulting sequencing reads as described previously [33]. Briefly, Illumina reads were trimmed and filtered to Q20 and a minimum length of 50 bp using BBDuk [34]. The paired reads were then merged using BBMerge with default settings [35]. The trimmed, filtered, and merged reads were then aligned to the Wuhan reference genome (NCBI GenBank accession NC_045512.2/MN908947.3) using BBMap with local alignment and a maximum insertion/deletion of 500 bp [34]. The amplicon primers were trimmed from sequences using align_trim from the ARTIC workflow/pipeline [36]. Consensus genomes were generated when possible, and single-nucleotide variants (SNVs) were determined using the SAMtools mpileup [37] and iVar (intrahost variant analysis of replicates) [38]. The resulting mappings were visualized and examined for artifacts, and when necessary, the genomes were manually closed in CLC Genomics Workbench v2021.0.4 (Qiagen; Valencia, CA, USA). The lineage determination of the consensus genomes was conducted using Pangolin (Phylogenetic Assignment of Named Global Outbreak LINeages; v.3.1.20) [39]. The clade assignments and consensus mutations were determined using Nextclade CLI 1.2.0 and Nextalign CLI 1.2.0. The viral genome data resulting from this study are available in GISAID, and their accessions are found in the Appendix A. Alignments were performed using MAFFT [40], and a Maximum Likelihood tree was generated with IQ Tree [41] ML (GTR+G) with 1000 bootstraps using a sub-set of the SARS-CoV-2 genomes available from the Global Initiative on Sharing All Influenza Data repository (GISAID accessed February 2022). The resulting trees were visualized and edited using FigTree (v1.4.4) [42]. The lineage distribution over time was visualized with a custom python script using the library Matplotlib [43]. Molecular docking was performed on the homology-modelled receptor-binding domain (RBD) of the SARS-CoV-2 spike (S) protein and human ACE2 proteins using the hybrid docking method of HDOCK (http://hdock.phys.hust.edu.cn/, accessed on 29 July 2022). The RBD protein sequences of the wild-type Wuhan virus (YP_009724390), A425, and A425 with mutated S494P were used as ligands, and PDB id 6LZG Chain A was chosen as the human ACE2 receptor sequence.

## 3. Results

### 3.1. Lineage Distribution Trends

A total of 795 SARS-CoV-2 positive samples were collected at Camp Lemonnier, Djibouti, between April 2020 and January 2022. Of those, 655 (82% of total samples) passed the library preparation quality control and thus proceeded for viral genome sequencing, resulting in 445 coding-complete genomes (68% of sequenced genomes) [44]. Of the remaining 210 consensus genomes, 144 genomes (22% of sequenced genomes) reached a consensus length ≥ 20,000 nucleotides (nt), while 66 samples (10% of sequenced genomes) either did not reach a consensus length ≥ 20,000 nt or were omitted due to sequence quality issues and thus were not included in subsequent analysis. An exception is the vaccine breakthrough infection (VBI) samples under 20,000 nt, which are discussed further below. The metadata (collection date, consensus genome length, Pango lineage, NextClade, and GISAID accessions) associated with 589 samples (the coding-complete and those with a consensus genome length ≥ 20,000 nt) can be found in Appendix A.

The Pango lineage assignments of the sequenced samples revealed trends of the emergence of specific lineages over the 21-month timeframe at Camp Lemonnier (Figure 1A,B). The samples collected between April 2020 and November 2020 were mainly of the B.1 lineage, except for June 2020, when there was a spike of samples assigned to the lineages B.1.324 (*n* = 11) and B.1.1.306 (*n* = 8). The 11 B.1.324 cases and 8 B.1.1.306 cases were from passengers on the same inbound flight from the U.S. From January 2021 through the beginning of February 2021, the predominant lineages were B.1.1.306, with another small cluster of B.1.324 cases. The small B.1.324 cluster occurred in individuals with a shared travel history, and all were collected on the same day in January. Mid-February 2021, B.1.2 emerged briefly, and then the predominant lineage was VOC B.1.1.7 (Alpha) from the end of February 2021 through mid-March 2021. The samples collected from mid-March 2021 through the end of May 2021 were predominantly VOC Beta. From July 2021 through November 2021, samples were primarily assigned VOC Delta and its sub-lineages. From December 2021 through January 2022, the VOC Omicron gained dominance, and the Omicron sub-lineage BA.1.1 was the most prevalent SARS-CoV-2 lineage at Camp Lemonnier. Within the Delta lineage samples, there were various sub-lineages represented, with each having fewer than 10 cases represented, except for the dominant sub-lineage, AY.127.1, which had 38 cases assigned (Figure 1B). Interestingly, despite its dominance at Camp Lemonnier at that time, AY.127.1 is a rare lineage, with only 337 genomes published to GISAID as of June 2022, accounting for less than 0.5% of lineages worldwide at that time. This therefore likely represents cluster outbreaks with two possible introductions occurring on Camp Lemonnier (February 2021 and August 2021). All 589 samples with lineages assigned are shown in a maximum likelihood phylogenetic tree, with the four main groups highlighted: VOCs Alpha, Beta, Delta, and Omicron (Figure 2).

### 3.2. Vaccine Breakthrough and Reinfection Cases

Of the 655 sequenced samples, 252 of these were VBI (either partial vaccination, full vaccination, or unknown vaccination details) cases (Figure 3): 80 with the Moderna vaccine, 77 with Pfizer-BioNTech, 68 with the Janssen (J&J) vaccine, 17 with AstraZeneca, 3 with Sputnik V, 1 with Sinovac, and 6 for which the information regarding vaccine type is missing. While the majority of these VBIs occurred in fully vaccinated individuals at least two weeks post-vaccination, two of the Moderna VBI cases tested positive after the first dose and prior to the second dose, classifying them as partial VBIs. Three (two Janssen and one Moderna) vaccinated cases tested positive prior to 14 days post-complete vaccination, also classifying as partial VBIs. Additionally, there are 51 cases with either one or more unknown vaccination dates, and thus vaccination status (fully or partial) is recorded as unknown. The remaining 196 of the VBIs were from fully vaccinated individuals (Table 1). A total of 45 cases of VBI were discovered from routine surveillance, as the individuals did not present with COVID symptoms, whereas 203 cases had symptoms consistent with COVID-19 [45]. The viral genome sequencing of the 252 VBI cases resulted in 211 coding complete genomes, 27 genomes ≥ 20,000 nt consensus genome length, and 14 samples that did not have adequate consensus length for Pangolin to assign a lineage (Table 1). Nearly all (237 of 238) of the VBIs with lineages assigned were confirmed to be VOCs and VUMs (variant under monitoring): 163 Omicron sub-lineages, 54 Delta lineages and sub-lineages, 17 Beta lineages, 2 Alpha lineages, and 1 Eta lineage (B.1.525) (Figure 3). All lineages except Eta were identified in this dataset as circulating at Camp Lemonnier during the timeframe these samples were collected (Figure 1a, Appendix A). The VBI cases followed general trends of VOC prevalence at Camp Lemonnier, with the first two VBIs collected assigned Alpha; starting 30 March 2021, Beta then became the dominant lineage identified from VBI cases. From May 2021 through November 2021, all viral genomes, regardless of the vaccination status of the patient, were assigned to the Delta and Delta sub-lineages. This timeframe also included four cases of confirmed reinfection. These data are consistent overall with the global dominance of Delta during this period. Starting in December of 2021, all VBIs were assigned to Omicron and its sub-lineages. Finally, the viral genome from one particular VBI, collected on 7 April 2021, was assigned to the B.1.1 lineage, a lineage circulating worldwide, but notable as this lineage accounts for only 1% of the total sequences published to GISAID as of June 2022.

### 3.3. Spike Mutations

In addition to characterizing the genomes by lineage, we also looked for specific mutations of interest and found that a group of 16 B.1 lineage viral genomes collected in November 2020 had the spike mutation of interest, S494P. It is striking since B.1 lineage viruses rarely have this mutation, which is reported in less than 0.5% of B.1 lineages viruses in GISAID even over a year after these samples were collected, as of June 2022. Located in the receptor-binding domain of the spike protein, a S494P mutation contributes to increased binding affinity to ACE2, the cellular receptor for SARS-CoV-2 [46], as well as reducing neutralizing antibody efficacy [47]. This group of samples likely represents an outbreak, with 14 of the samples collected on the same day, 16 November 2020, and is further supported by the clustering of these genomes on the phylogenetic tree (Figure 4A). S494P was also found in a cluster of nine BA.1.1 (Omicron) lineage viruses collected in December 2021, all of which were VBIs (Figure 4B). Similarly to B.1, this mutation is rare in this lineage, being found in less than 0.5% of BA.1.1 lineage virus sequences as of June 2022. Consistent with another study [46] showing increased binding affinity between S494P to human ACE2, our analysis with HDOCK showed a predicted increased binding affinity of a BA.1.1 lineage genome (sample A425) with the addition of the S494P spike mutation. The docking score decreased to −364 from −347 with the addition of S494P.

Additionally, six genomes of the B.1.1.306 lineage have another spike mutation of interest not commonly associated with this lineage, P681H (Appendix A). P681H is suggested to increase infectivity of the virus as it directly neighbors the furin cleavage site of the spike protein [48]. One sample was assigned to lineage B.1.1.25 but also had the P681R spike mutation, a mutation found in less than 0.5% of the B.1.1.25 lineage viruses on GISAID as of June 2022. Similar to P681H, P681R may increase infectivity of the virus [50].

In addition to the BA.1.1 VBI samples with spike S494P discussed above, we noted other viral genomes from VBIs in this study having mutations of interest that are not commonly associated with their assigned lineages. One genome from a VBI case, which was assigned lineage B.1.1, had the spike mutations of concern, E484K and N501Y, which are only found in less than 0.5% of the published B.1.1 lineage sequences as of June 2022 and are known to reduce antibody neutralization efficacy alone (E484K) or in combination (E484K/N501Y) [49]. This particular genome also had the spike mutation H1271Y, which is present in the C-terminus of the S2 intraviron region of the spike protein and is involved in the incorporation of the spike protein into the virion and the spike-mediated virus–host cell membrane fusion [50]. Strikingly, H1271Y is present in only one other B.1.1 sequence published to GISAID, and that was collected in Switzerland.

Two other genomes, both from VBIs and assigned lineage B.1.1.7 (Alpha), had the spike mutation H49Y, a mutation found in less than 0.5% of the Alpha (B.1.1.7) sequences on GISAID as of June 2022. It was also found in 20 other samples of the VOC Alpha genomes sequenced in this dataset that were not VBIs. H49Y is in the N-terminal domain of the S1 sub-unit of the spike protein and has been shown to increase the stability of the spike protein and to increase cellular entry [46,51]. Due to the rarity of this mutation in the B.1.1.7 lineage sequences, this group likely represents a cluster outbreak occurring at Camp Lemonnier, with the first case collected on 25 February 2021 and the last case collected on 10 March 2021, further supported by their clustering on the phylogenetic tree (Figure 4C).

## 4. Discussion

An analysis of 795 samples from Camp Lemonnier, Djibouti, from April 2020 through January 2022 resulted in the detection of VOCs and VUMs, the identification of mutations of interest in lineages not normally associated with them, the identification of vaccine breakthrough infections, and the general knowledge of circulating lineages and mutations in this geographic location. Overall, during the timeframe of this study, Camp Lemonnier followed global trends for the circulation of certain lineages, such as B.1, B.1.2, Alpha, Delta, and Omicron. Lineages B.1.1.306 and B.1.324, both having small clusters of infection in this study, are rarer lineages; as of June 2022, B.1.1.306 comprised less than 0.5% of worldwide lineages, with the greatest prevalence in Zambia and then Djibouti [14]. B.1.324 comprised less than 0.5% of worldwide lineages as of June 2022, with the greatest prevalence in Djibouti (including all samples discussed in this text) and the British Virgin Islands [14]. The samples in this dataset with these lineages likely represent small cluster outbreaks, which is further supported by the 11 cases of B.1.324 being from passengers arriving from the U.S. together on an inbound flight. The most prevalent lineage circulating in the dataset is BA.1.1, a sub-lineage of Omicron; within the vaccine breakthrough samples, VOC Omicron is the predominant lineage, followed by the VOC Delta. The VOCs Delta, Beta, and Omicron were documented to partially evade the immunity induced by the currently available vaccines [25,26,30,52], which is consistent with the reduced efficacy of antibodies to Beta, Delta, and Omicron due to the mutations present in the spike protein [8,9,10,11,25,29,53].

In addition to determining the circulating lineages at Camp Lemonnier, we identified spike mutations of interest in lineages where they are rarely found. This included a group of samples from the B.1 and BA.1.1 (Omicron) lineages with the spike S494P mutation, a mutation of interest as it reduces neutralizing antibody efficacy [47]. Other spike mutations of interest identified in lineages where they are rarely found included P681R, P681H, E484K, N501Y, and H49Y. These all are associated with previous evidence in the literature of increasing infectivity and/or reduced neutralization by antibodies [46,48,49,54]. Although the major variants circulating at Camp Lemonnier broadly appear to align with the variants that circulated in other regions of the world at the same time, the finding of multiple “rare” mutations in these different lineages at Camp Lemonnier, in a region of the world where relatively few SARS-CoV-2 genomic sequences have been reported overall, suggests that uneven sampling from various geographic regions may be skewing the databases toward more industrialized areas and thereby missing or underrepresenting the viral genetic variations in the rest of the world. Tracking emerging mutations of interest in new or even previously described lineages is critical for advising on therapeutics and public health policies, as well as for the prediction of risk to deployed military forces around the globe.

The viral sequences in this analysis were the first published to GISAID from the country of Djibouti. Despite some unique characteristics of infectious disease surveillance in a deployed military setting, it has been demonstrated previously in influenza-like illness (ILI) surveillance that more than a quarter of the ILI cases surveilled at Camp Lemonnier were among the personnel living and interacting within the local community [55]. Therefore, it is reasonable to infer that the SARS-CoV-2 surveillance data from Camp Lemonnier would include not only lineages that may be introduced from recent personnel movement but also lineages that are circulating regionally within Djibouti. Overall, this dataset provides a critical addition to the knowledge base of SARS-CoV-2 lineages circulating in Camp Lemonnier, Djibouti, specifically and perhaps in Africa in general, and it underscores the importance of efforts aimed at sampling from various geographic regions. The surveillance of SARS-CoV-2 infections and vaccine breakthroughs in the U.S. military and adjunct staff at Camp Lemonnier is necessary for the implementation of control measures, as well as for advising policy on protective measures for the overseas military.

## Figures and Tables

**Figure 1 viruses-14-01918-f001:**
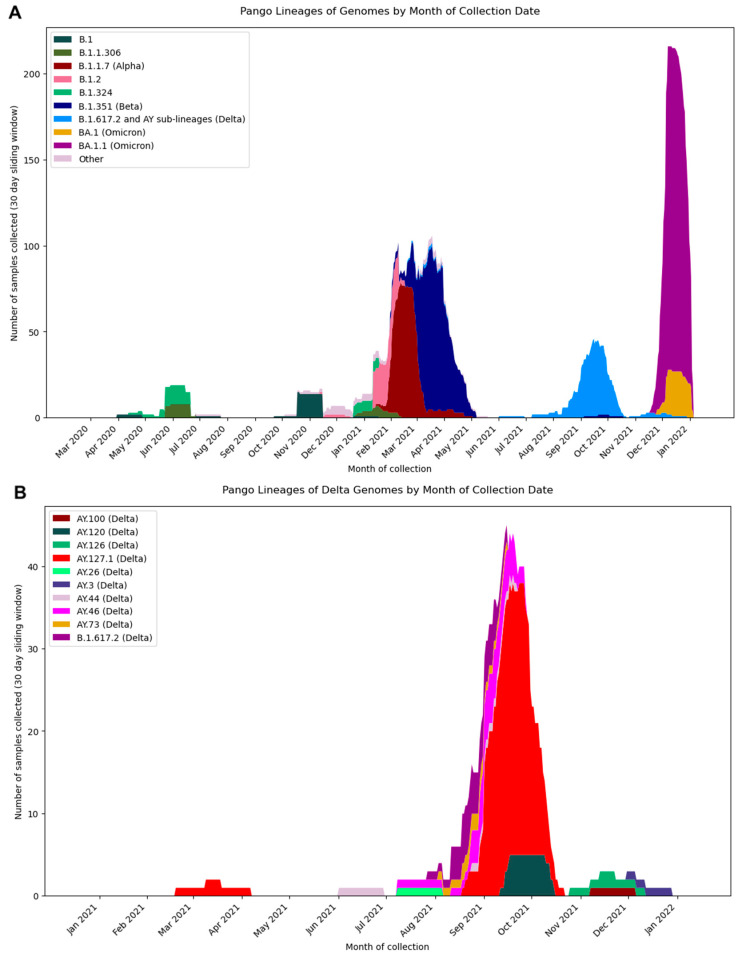
Pango lineage distributions over time: (**A**) the entire dataset of genomes ≥ 20,000 nt collected at Camp Lemonnier, Djibouti; (**B**) the genomes assigned to Delta (B.1.617.2 and AY sub-lineages).

**Figure 2 viruses-14-01918-f002:**
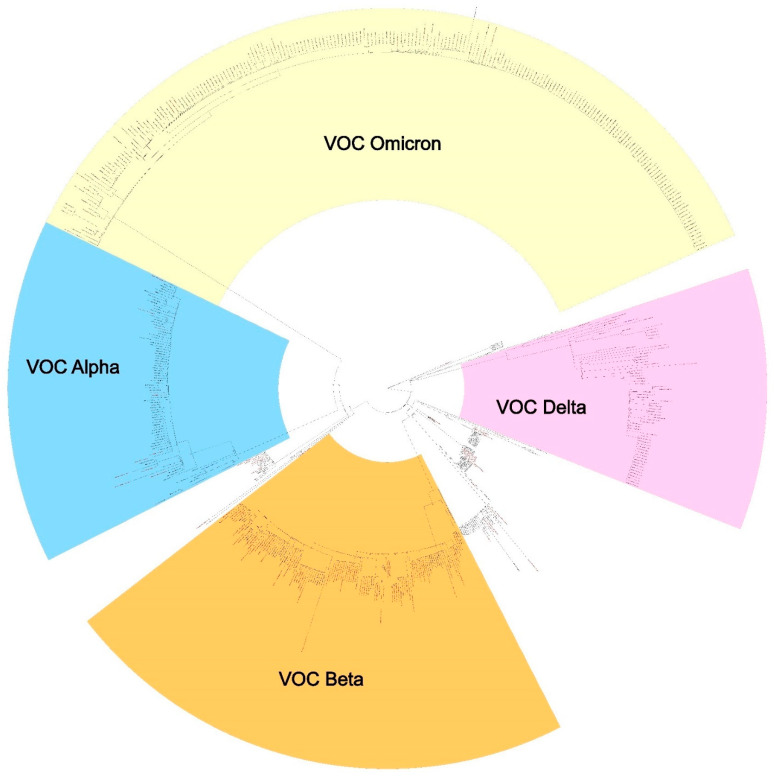
Maximum likelihood tree of all samples > 20,000 nt with their assigned Pango lineages. The Alpha lineage samples are highlighted blue, Beta are yellow-orange, Delta are purple, and Omicron are light yellow. The coding-complete genomes are shown in black, while the non-coding-complete consensus genomes are shown in red. Node labels show bootstraps.

**Figure 3 viruses-14-01918-f003:**
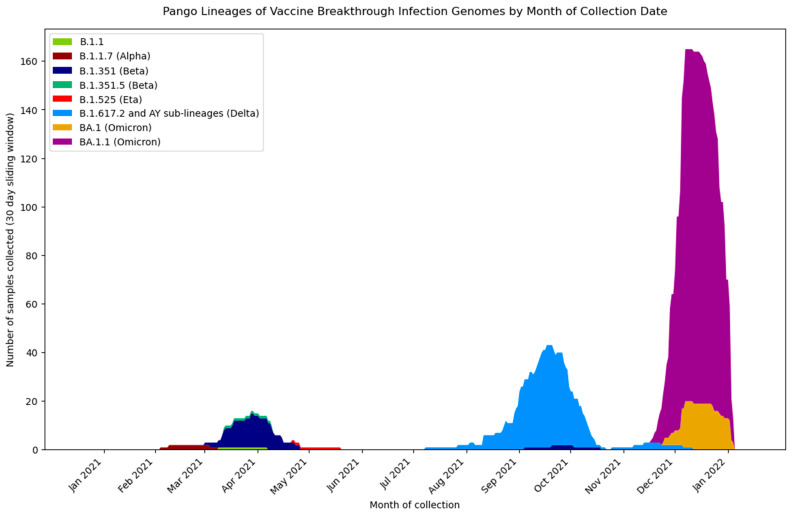
Pango lineage distributions over time of vaccine breakthrough infection genomes, excluding genomes < 20,000 nt.

**Figure 4 viruses-14-01918-f004:**
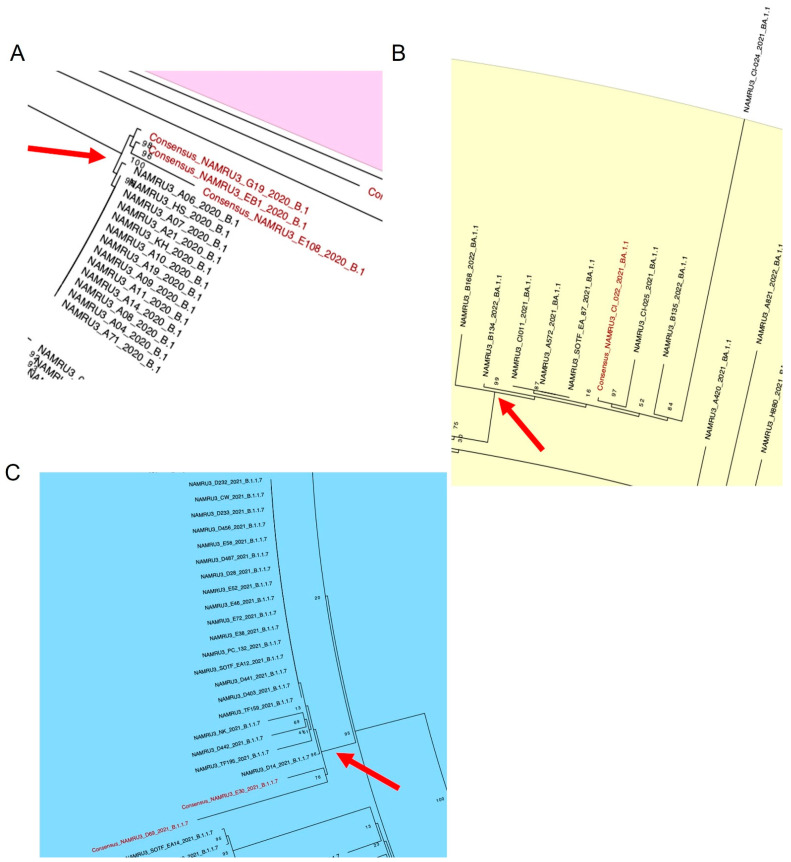
Close-up of phylogenetic clusters of samples with rare spike mutations from Figure 2: (**A**) 16 samples of lineage B.1 with spike mutation S494P; (**B**) 9 samples of lineage BA.1.1 with spike mutation S494P; (**C**) 22 samples of B.1.1.7 with spike mutation H49Y. The coding-complete genomes are shown in black, while the non-coding-complete consensus genomes are shown in red. Nodes are labeled with bootstraps.

**Table 1 viruses-14-01918-t001:** Summary of vaccine breakthrough, partial breakthrough, unknown vaccination status, and unvaccinated cases in dataset.

Vaccination Status	Vaccine (#)	Genomes	Pango Lineages
Vaccine breakthrough infection	Pfizer (60)	48 (80%) coding complete, 10 (17%) ≥ 20,000 nt, 2 (3%) < 20,000 nt	1 (2%) Beta, 7 (12%) Delta, 50 (83%) Omicron, 2 (3%) no lineage assigned
	Moderna (64)	60 (94%) coding complete, 3 (5%) ≥ 20,000 nt, 1 (2%) < 20,000 nt	2 (3%) Beta, 8 (12%) Delta, 53 (83%) Omicron, 1 (2%) no lineage assigned
	J&J (61)	44 (72%) coding complete, 10 (16%) ≥ 20,000 nt, 7 (11%) < 20,000 nt	11 (18%) Beta, 16 (26%) Delta, 25 (41%) Omicron, 2 (3%) other, 7 (11%) no lineage assigned
	Other/unknown (11)	10 (91%) coding complete, 1 (9%) ≥ 20,000 nt	1 (9%) Delta, 10 (91%) Omicron
Partially vaccinated	Pfizer (0)	-	-
	Moderna (3)	2 (67%) coding complete, 1 (33%) ≥ 20,000 nt	2 (67%) Alpha, 1 (33%) Omicron
	J&J (2)	1 (50%) ≥ 20,000 nt, 1 (50%) < 20,000 nt	1 (50%) Beta, 1 (50%) no lineage assigned
	Other/unknown (0)	-	-
Unknown vaccination level (partial or full)	Pfizer (17)	16 (94%) coding complete, 1 (6%) < 20,000 nt	7 (41%) Delta, 9 (53%) Omicron, 1 (6%) no lineage assigned
	Moderna (13)	11 (85%) coding complete, 1 (8%) ≥ 20,000 nt, 1 (8%) < 20,000 nt	1 (8%) Beta, 1 (8%) Delta, 10 (77%) Omicron, 1 (8%) no lineage assigned
	J&J (5)	4 (80%) coding complete, 1 (20%) < 20,000 nt	4 (80%) Omicron, 1 (20%) no lineage assigned
	Other/unknown (16)	16 (100%) coding complete	1 (6%) Beta, 14 (88%) Delta, 1 (6%) Omicron
Unvaccinated	N/A (351)	235 (67%) coding complete, 116 (33%) ≥ 20,000 nt	84 (24%) Alpha, 113 (32%) Beta, 11 (3%) Delta, 51 (15%) Omicron, 92 (26%) other

Note: Consensus genomes less than 20,000 nucleotides (nt) in length from unvaccinated individuals were not included in the analysis.

## Data Availability

The published genomes are found on GISAID; see Appendix A for accessions.

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
