# Peer review of "SARS-CoV-2 Infections in Vaccinated and Unvaccinated Populations in Camp Lemonnier, Djibouti, from April 2020 to January 2022"

_viruses, 2022, doi:10.3390/v14091918_

Round 1
Reviewer 1 Report
Summary:
Dr. Arnold et al. wrote a descriptive study of SARS-CoV-2 infections variants lineage in Camp Lemonnier, Djibouti. Analysis is clear and well written and span over 21 month (April 2020 to January 2022). Beyond lineage, authors also describe breakthrough infections and potential spike mutation emerging. My main concern represent the claim regarding the study being descriptive of the under-sampled country of Djibouti with the authors highlighting the issue of under-represented third world countries.
Major:
- Is Camp Lemonnier a US base with US military personnel only? If that is the case, how representative of Djibouti outbreak are the sequences? I am curious as to if these cases were introduced by newly deployed personnel on the base and subsequent community exposures, or rather truly directly related to local population exposure.
Minor
- Line 89: Typo: Pfizer instead of "Pfizet".
- Figure 2 appears to have a low resolution in the manuscript submitted. Authors need to make sure the resolution in the final manuscript is better.
Author Response
Dr. Arnold et al. wrote a descriptive study of SARS-CoV-2 infections variants lineage in Camp Lemonnier, Djibouti. Analysis is clear and well written and span over 21 month (April 2020 to January 2022). Beyond lineage, authors also describe breakthrough infections and potential spike mutation emerging. My main concern represent the claim regarding the study being descriptive of the under-sampled country of Djibouti with the authors highlighting the issue of under-represented third world countries.
Major:
- Is Camp Lemonnier a US base with US military personnel only? If that is the case, how representative of Djibouti outbreak are the sequences? I am curious as to if these cases were introduced by newly deployed personnel on the base and subsequent community exposures, or rather truly directly related to local population exposure.
We have added additional text in the discussion to clarify this point (lines 347-353 of the tracked changes version).
Minor
- Line 89: Typo: Pfizer instead of "Pfizet".
Thanks for finding this, we have corrected it.
- Figure 2 appears to have a low resolution in the manuscript submitted. Authors need to make sure the resolution in the final manuscript is better.
The Microsoft Word document compressed the image and a full resolution image will be provided for publication. We have updated the resolution in the document as well.
Reviewer 2 Report
The manuscript deals with study of SARS-CoV-2 infections in vaccinated and unvaccinated populations in Camp Lemonnier, Djibouti from April 2020 to January 2022. The manuscript is interesting, in my opinion, it should be revised.
Comments,
1) Abstract should be revised.
2) In introduction section, recent works should be discussed and cited, Chemical Engineering Journal 420, 127575, 2021; Chemical Engineering Journal 414, 128759, 2021; ACS applied bio materials 4 (2), 1178-1190, 2021
3) 2.2. RNA extraction and genome sequencing, can be as flow chart.
4) 3.1. Lineage Distribution Trends should be explain in detail.
5) Figures quality is low.
6) Conclusions should be rewritten.
7) Reference section should be updated.
8) There are some typo and English errors, it should be rectified.
Author Response
The manuscript deals with study of SARS-CoV-2 infections in vaccinated and unvaccinated populations in Camp Lemonnier, Djibouti from April 2020 to January 2022. The manuscript is interesting, in my opinion, it should be revised.
Comments,
- Abstract should be revised.
Abstract has been revised for clarity.
- In introduction section, recent works should be discussed and cited, Chemical Engineering Journal 420, 127575, 2021; Chemical Engineering Journal 414, 128759, 2021; ACS applied bio materials 4 (2), 1178-1190, 2021
An introduction into diagnostics for SARS-CoV-2 has been added and citations regarding nanosensors and PCR have been included (lines 56-65 of the tracked changes version of the revised manuscript).
- 2. RNA extraction and genome sequencing, can be as flow chart.
Similar to other genomic surveillance studies, the methods of viral genome sequencing are described and we believe a flow chart is beyond the scope of this paper.
- 1. Lineage Distribution Trends should be explain in detail.
Additional descriptions of the lineage trends seen in the Delta sub-lineages have been added.
- Figures quality is low.
This is due to the compression of the figures in a Microsoft Word document, full resolution images will be provided for publication. We have updated Figures 2 and 4 in the text to be higher resolution as well.
- Conclusions should be rewritten.
In response to reviewer 1 as well, we clarified our conclusions and updated the text to reiterate the impact for surveillance in Djibouti.
- Reference section should be updated.
Additional references were added for the introduction and discussion.
- There are some typo and English errors, it should be rectified.
An additional round of the edits were made to rectify any grammatical errors.
Reviewer 3 Report
This is a useful interesting topic. Reading the paper showed well organized steps that reflect great experience and efforts.
Minor modifications are suggested such as:
|
I prefer docking procedure at specialized accurate programs more than websites. (not mandatory) |
Author Response
This is a useful interesting topic. Reading the paper showed well organized steps that reflect great experience and efforts.
Minor modifications are suggested such as:
|
I prefer docking procedure at specialized accurate programs more than websites. (not mandatory) |
HDOCK also runs as a standalone program and we have achieved similar results using HDOCK when compared with analyses involving molecular modeling and molecular dynamic simulations using Modeller and GROMACS.
Round 2
Reviewer 1 Report
Authors made good effort to address my comments.